

# A comprehensive approach for waste management with GAN-augmented classification

Yashashree Mahale[1], Nida Khan[1], Kunal Kulkarni[1], Shilpa Gite[1,2], Biswajeet Pradhan[3], Abdullah Alamri[4], Chang-Wook Lee[5], Nandhini K.[2] and Mrinal Bachute[2]

[1] Department of Artificial Intelligence and Machine Learning, Symbiosis International Deemed University, Symbiosis Institute of Technology, Pune, Maharashtra, India
[2] Symbiosis Centre for Applied Artificial Intelligence, Symbiosis International Deemed University, Symbiosis Institute of Technology, Pune, Maharashtra, India
[3] Centre for Advanced Modelling and Geospatial Information Systems (CAMGIS), University of Technology Sydney, Sydney, New South Wales, Australia
[4] Department of Geology and Geophysics, College of Science, King Saud University, Riyadh, Saudi Arabia
[5] Department of Science Education, Kangwon National University, Chuncheon-si, Republic of South Korea

Corresponding authors
Biswajeet Pradhan,
biswajeet24@gmail.com
Chang-Wook Lee,
cwlee@kangwon.ac.kr

## ABSTRACT

Image processing and computer vision highly rely on data augmentation in machine learning models to increase the diversity and variability within training datasets for better performance. One of the most promising and widely used applications of data augmentation is in classifying waste object images. This research focuses on augmenting waste object images with generative adversarial networks (GANS). Here deep convolutional GAN (DCGAN), an extension of GAN is utilized, which uses convolutional and convolutional-transpose layers for better image generation. This approach helps generate realism and variability in images. Furthermore, object detection and classification techniques are used. By utilizing ensemble learning techniques with *DenseNet121, ConvNext, and Resnet101*, the network can accurately identify and classify waste objects in images, thereby contributing to improved waste management practices and environmental sustainability. With ensemble learning, a notable accuracy of 99.80% was achieved. Thus, by investigating the effectiveness of these models in conjunction with data augmentation techniques, this novel approach of GAN-based augmentation cooperated with ensemble models aims to provide valuable insights into optimizing waste object identification processes for real-world applications. Future work will focus on better data augmentation methods with other types of GANS architectures and introducing multimodal sources of data to further increase the performance of the classification and detection models.

## INTRODUCTION

Recently, with the increased interest in environmental problems around the world, more and more people have sought ways to be more sustainable in almost every aspect.

An important focus is better management of waste to maintain a healthy environment. With growing concerns about sustainability, accurate waste management has become crucial. As such, we find a shift away from the old ways of managing waste to the use of smart computer systems, including artificial intelligence (AI), which make the process much faster and easier (*Karn et al., 2023*). It assists in the rapid and accurate sorting of various kinds of waste, making modern waste management much better. It has become the most famous automated waste sorting device because of its capability in processing garbage waste classification with minimum human error (*Rutqvist, Kleyko & Blomstedt, 2019*). Waste disposal and management have become an enormous social issue and a major cause of environmental deterioration with special emphasis on pollution of the environment and habitat destruction, greenhouse effect and global warming among others. The conventional ways of sorting the waste involve sorting the waste using the help of workers, and this takes a lot of time besides not being very accurate. In addition, with sustained advancement in urbanization, the amount of waste produced increases; this exerting significant strain on current waste treatment infrastructure. In order to address these problems, the existence of automated, accurate and efficient waste classification systems that would facilitate the sorting process in support of the sustainable waste management is highly demanded (*Cai et al., 2022*; *Arbeláez-Estrada et al., 2021*). In this regard, the quality and volume of training data available play a significant role in determining how effective these automated systems are. A lack of robust training data, characterized by limited size and diversity, remains a significant barrier to accurate waste material categorization. Limited accuracy in waste categorization algorithms has often been compromised by incomplete and restricted datasets, and this has led to not-so-perfect sorting results, which in turn remain a source of raising environmental concerns. As a result, automated systems may find it difficult to differentiate between the different categories of waste materials accurately, and this may result in sorting mistakes and inefficiencies. This work is therefore prompted by the need to meet these challenges employing current advancements in computer vision and deep learning as the foundation of sustainable and efficient waste management.

Recognizing these challenges, this study proposes a new way of improving the waste object classification that uses methods of synthetic data augmentation, in particular, the generative adversarial networks (GANs) is used because traditional way of data gathering and augmentation methods might not be able to keep up with changes in the waste materials situation. In fact, by learning from the data distribution, GANs identify subtleties and complex patterns similar to those seen in the training data, unlike the conventional generative models which rely on explicit probability distributions (*Cai et al., 2022*). As this adversarial dynamic pushes the generator towards producing increasingly realistic samples, the convergence is achieved. Further, this dynamic produces synthetic data that closely resembles the distribution of real-world data. GANs offer a good option to add artificial samples to the training datasets in the context of data augmentation. GANs may create new instances of data with realistic variations and nuances by exploiting the latent space that they have learned in the process of training (*Hsia et al., 2022*). Data augmentation is an important method of image processing which is mainly used for

increasing the generalization performance of the model. Integrating such techniques expands the sizes of the datasets in image processing models so that they can accurately recognize and classify images.

The advantages of using the EfficientNet, ResNet50, and VGG16 models for image classification are multifaceted and enormous because of the complexity of tasks related to image analysis and recognition (*Jaikumar, Vandaele & Ojha, 2020*). This, coupled with the fact that applications targeting specific uses, such as waste bottle segmentation and defect detection in wind turbine blades, have shown that enhanced deep learning pipelines excel in localization and segmentation, surpassing their original counterparts. However, this does not mean that the models aren't versatile or not useful in other applications apart from conventional image classification (*Li et al., 2021*). Indeed, the versatility and effectiveness of the models have succeeded in interdisciplinary applications, such as the sorting process in wastes and recyclables detection, not only in environmental concerns but in resource management as well.

Overall, GANs represent an extremely potent tool for data augmentation and generative modelling, enabling unparalleled abilities to produce realistic data in various domains. This research uses generative learning's complex data augmentation techniques in trying to solve the critical requirement of improved waste management procedures (*Karn et al., 2023*). In this regard, our purpose is to mitigate the consequences of poor training data through synthetic data generated by GANs and, therefore, develop waste sorting systems that are more accurate, efficient, and sustainable. The research article concentrates on data augmentation techniques and their employment in waste management domain with deep learning techniques. The utilization of ResNet50 and VGG16 models in image classification reveals many advantages in specific applications such as image classification, particularly in complex image sensing and image recognition (*TensorFlow, 2022*). Applications related to waste bottle segmentation and wind turbine blade defect detection have exposed the enhanced capabilities of enhanced deep learning pipelines in the localization and segmentation of objects far more efficiently and effectively compared to their original versions.

This research makes several significant contributions to the field of waste object classification. Firstly, it adjusts the training procedure of GANs to prioritize the creation of synthetic data for underrepresented or challenging waste categories within the original dataset. This adaptation ensures that the model is trained on a more comprehensive range of waste materials, enhancing its ability to accurately classify diverse objects. Secondly, the study emphasizes the generation of artificial waste object images that closely mimic real-world variations, thereby expanding and diversifying the dataset. By incorporating GAN-generated synthetic images into the training dataset, the robustness and generalization of the classification model are significantly increased, leading to improved performance in real-world scenarios.

This research provides a number of primary contributions to the study of waste object classification. Firstly, it adapts the Deep Convolutional GAN (DCGAN) framework to focus on the synthesis of virtual underrepresented categories of waste, improving the number and quality of training data sets. Traditional approaches to augmentation suffer

from the lack of incorporating such variations; this work presents an effective GAN based method that augments based on the data distribution and domain specific characteristics hence providing better generalization. Secondly, the study also proposes an innovative method to form an ensemble of DenseNet121, ConvNext, and ResNet101 to achieve the optimized features for classification. Collectively, these works show the viability of integrating state-of-the-art augmentation methods with novel detection-classification workflows that can support the future augmentation of intelligent waste management systems.

Therefore, the development of approaches like GANs in deep learning provides an incredible chance to emerge out of these complications. Contrary to that, GANs are capable of synthesizing new synthetic data to mitigate data limitation and variability, whereas the Residual Network (ResNet), Visual Geometry Group (VGG), Inception model offer immense class distinction precision in complex surroundings. This work is driven by these emerging technologies and the potential they hold for the future of waste management through deployment of self-organizing scalable automated systems. Lastly, the research deploys the trained classification model into a practical waste management application, enabling real-time classification in various environmental settings. This practical implementation demonstrates the scalability and accuracy of the proposed approach, highlighting its potential for addressing challenges in waste management systems.

## RELATED WORK

A comprehensive analysis of the research in data augmentation using GANs and various computer vision-based classification algorithms are analysed in this section.

In *Chatterjee et al. (2022)*, the authors investigated methods to improve image classification by utilizing a novel lightweight GAN model for creating synthetic image data from South Korean plastic industry bottle images. Additionally, pre-trained ImageNet models were optimized for the classification of plastic bottles. The proposed Inception-Ensemble model achieved an impressive accuracy of 99%. The study suggests future analysis with big data and exploration of trash management apps and datasets. *Kumsetty et al. (2023)* focused on classification of waste with data augmentation and transfer learning models on famous datasets for waste classification such as TrashNet and Trash Annotations in Context (TACO). The TrashNet dataset is employed in the classification of waste and is comprised of images of wastes with six classes of labels. Likewise, TACO provides synchronized waste images with high-quality annotations in real scenarios, which benefits the fine-tuning for the complex classification tasks. There are also problems of diversity and representation in both datasets, where for the less represented categories there are problems solved by synthetic data augmentation. The authors employed synthetic image generation techniques using GANs and utilized pre-trained models like ResNet-34, VGG-16, and ResNet-101 (*Abbas & Singh, 2018*). Using a benchmark dataset that included both overlapping as well as non-overlapping images, the method's accuracy was 93.13%. However, no noteworthy findings were found for artificial datasets created for medical waste classes using GANs.

The authors in *Situ et al. (2021)* investigated the use of Style-Based Generative Adversarial Networks (StyleGANs) for identifying picture defects. The study used popular convolutional neural network (CNN) classifiers and concentrated on captured photos. For the various cases, the mean Average Precision (mAP) values were 94%, 92.5%, 91.7%, 92.1%, and 90%. Additionally, the study looked into the adjustment of hyperparameters and complexity of the model. In general, the finding of flaws automatically based on StyleGANs and CNN classifiers has presented bright prospects.

In the study (*Poudel & Poudyal, 2022*), CNNs were applied to the classification of images in waste management. Models which were pre-trained such as ResNet50 and VGG16 were used for training models on a waste dataset with seven types of waste such as trash, cardboard, glass, metal, organic, paper, and plastic. Among the models, InceptionV3 was the most effective, and VGG19 had a lower accuracy. In general, the results have proved that CNNs are capable of material classification in waste. Among the models, ResNet18 and ResNet50 gained very high validation accuracy, while VGG16, in comparison, had less accuracy. The results were very promising and showed that CNNs could be used for the classification of waste material. *Alalibo & Nwazor (2023)* compared the performance of ResNet50, ResNet18, and VGG16 CNN models in the TrashNet dataset was compared. The highest validation accuracy of 87.8% was obtained by ResNet18 after fine-tuning. The approach focuses on the classification of household waste types from images.

The approach proposed in *Gupta et al. (2022)* uses both CNNs and the Capuchin Search Algorithm to combine these features and implements a learning model which was hybrid that uses Error-Correcting Output Code (ECOC) with artificial neural networks (ANN). Here a new approach for deep learning is used to automatically sort household waste images. Results are improved to achieve higher accuracy rates compared to existing methods, registering results of 98.81% and 99.01% on respective databases, which posits the feasibility of highly accurate and efficient automation for waste classification with reduced errors and environmental hazards.

In this context, this article proposed the smart dustbin system (SDS) as a solution to address the limitations of traditional waste management methods, which are often costly and inefficient, leading to sanitation problems and health hazards (*Liu, Zhao & Sun, 2017*). Leveraging the latest advancements in deep learning, computer vision, and Internet of Things (IoT), the technological backbone of SDS ensures real-time monitoring and optimization of waste collection processes (*Arthur, Shoba & Pandey, 2024*). SDS is tailored for key urban settings such as universities, malls, and high-rise buildings, aiming to reduce labour costs and improve cleanliness. Through an in-depth literature review, recent research discussions, and comparisons with existing methods, this article outlines the viability of the SDS methodology in transforming waste management practices (*Rahman et al., 2024*). The "BDWaste" dataset comprises 2,497 meticulously captured images representing two main waste types: digestible and indigestible. These waste types encompass 10 different categories, with images captured in both indoor and outdoor settings. Notably, the dataset excludes blurry or noisy images, ensuring high quality. The study employs a convolutional neural network model pretrained on MobileNetV2 and

InceptionV3, achieving impressive classification accuracies of 96.70% for indigestible waste and 99.70% for digestible waste.

In *Alsubaei, Al-Wesabi & Hilal (2022)*, authors presented Deep Learning-Based Small Object Detection and Classification for Garbage Waste Management (DLSODC-GWM), a modified deep learning techniques for detecting and classifying small object during managing waste. They optimized an improved RefineNet model using an arithmetic optimization algorithm (AOA) for object detection with employing a functional link neural network (FLNN) for waste classification. The authors in *Alsubaei, Al-Wesabi & Hilal (2022)* evaluated on benchmark datasets shows promising results, with a maximum accuracy of 98.61%, surpassing existing methods. Another study is a comprehensive overview of the latest developments of visual object detection leveraging deep learning techniques. It categorizes the discussion in three main sections: learning strategies, applications & benchmarks and detection components. Multiple factors influence the performance of detection, such as architecture of the detector and feature learning, are analysed in detail.

The authors in *Arbeláez-Estrada et al. (2021)* conducted a comprehensive systematic review of waste identification methods in automated systems, highlighting the importance of machine learning and sensor-based techniques, especially under limited data conditions. Their review emphasizes the challenge of variability in waste appearance and the growing reliance on data augmentation and deep learning to address these challenges.

The related work focuses on some of the most important achievements in waste classification and management using deep learning, computer vision, and generative models (*Peng et al., 2020*). However, the related work also appreciates the ongoing utilization of multimodal dataset to enhance waste classification's progress. Although the methods using TrashNet and TACO are still common, studies such as BDWaste indicate the importance of the variability of the images' capture and detailed labels. The images in the TrashNet dataset, which is popular in articles related to waste classification, are well-tagged by category and have been useful in comparing classification models. Likewise, TACO provides real pictures of trashes with rather detailed annotations which could be help for certain fine-tuned classification studies. But even these datasets do not capture new changes in categorization of wastes for instance, inclusion of medical waste or mixed material items. There are also problems of diversity and representation in both datasets, where for the less represented categories there are problems solved by synthetic data augmentation. Specifically, next-generation GAN architectures for synthetic data generation present a plausible method to resolve these gaps by increasing the diversity of datasets while training models for classification. Researches have used methods like lightweight GANs for generating synthetic data, and pre-trained models including ResNet, VGG, and Inception for waste categorization, and StyleGAN for recognizing defects. Specifically, using I-Ensemble models for small object detection the accuracy of at 99% and with DLSODC-GWM at 98.61% shows the efficiency of these technique. Combining methods such as CNNs, ANN and ECOC have been extended even further to attain accuracy rates greater than 99%. Challenges posed by waste management coupled with the concept of the IoT and MobileNetV2 as illustrated by smart dustbin systems (SDS) present

**Table 1 Summary of literature survey.**

| Reference | Publication year | Dataset used | Methodology | Results | Future scope/Gaps |
|---|---|---|---|---|---|
| Ahtesham (2024) | 2024 | Waste Classification Data from Kaggle | GANS architecture and MobileNetV2 architecture was used for image generation and classification respectively. | Accuracy of 51% and the weighted average F1-score is 0.50. | Data augmentation techniques and ensemble learning can be combined for better predictions. |
| Bird et al. (2022) | 2022 | Dataset of lemon from SoftwareMill | A conditional GAN is utilized and trained for 500 epochs | Accuracy-16%, effective double production at a loss of 7.59% accuracy. | Need for image segmentation and quality of dataset. |
| Alsabei et al. (2021) | 2023 | StyleGan with ResNet50, InceptionV3, Xception and VGG16 with GDA | The models with ResNet50, InceptionV3, Xception and VGG16 with varying GDA | Each model was trained with six models with Ratio GDA with best accuracy being 95%, 87%, 94%, 9 0%, 96%, 93% respectively. | More data is needed for better training of GAN to avoid noise |
| Zhang et al. (2024) | 2024 | Coal and gangue augmented samples | Dual attention deep convolutional generative adversarial network (DADCGAN), YOLOv4 model | DADCGAN achieved lowest FID, YOLOv4-92.4% accuracy | Apply and compare other computer vision models |
| Kundu, Sharma & Pillai (2024) | 2024 | - | Computer vision | Accuracy-95.6% | Apply advanced artificial learning |
| Shi et al. (2023) | 2023 | Constructed RTrash dataset | SIM-YOLOv7 algorithm | SIM-YOLOv7 provided better result than YOLOv7 | Comparative analysis with various other models can be performed. |
| Erabati & Araujo (2024) | 2024 | Waymo and KITTI datasets | LiDAR-based 3D object detection, Deep and Light-weight Voxel Transformer (DeLiVoTr) | Inference speed−20.5 FPS | The model can be applied in other domains. |

an opportunity for improving waste management in urban areas. However, issues regarding the efficiency of artificial dataset still remain and particular type of waste such as medical type of waste and there is need for more analysis using big data and better generalization of results in future studies.

The literature study conducted for the mentioned research is depicted in Table 1 above. Various types of GANs models were analysed for different applications of data augmentation across domains such as healthcare, material science, waste objects, agricultural products images, *etc*. Our exploration encompassed the analysis of different hyperparameter tuning and optimizations applied to the CNN layers within GANs architecture for image data augmentation and classification. Additionally, several machine vision techniques for image processing and classification were explored. The literature also elucidated the limitations stemming from the scarcity of image datasets and the necessity for data. The limited number of waste images in the dataset have been mentioned by

several research articles, resulting into lower accuracy of the classification models and hence need of data augmentation is encouraged. Also, several researches aim to apply, compare and analyse several models for data augmentation as well as classification. Further research can be done for other domains too using similar approach and then comparative analysis can be made.

# DATA AND METHODOLOGY

## Dataset description

For this study, the dataset for waste of various categories was used (*CCHANG, 2018*). This dataset has been collected from the Kaggle repository. This dataset contains images of different types of waste materials, labelled into six classes: cardboard, biological waste, plastic, paper, trash, and metal. These categories represent the diverse range of waste encountered in daily life. The dataset was meticulously curated to include high-resolution images captured under various environmental conditions and settings. The inclusion of a diverse range of waste classes ensures comprehensive training and model evaluation, thereby enhancing the robustness and generalization of the model across different waste types.

The dataset contains images of waste materials divided into six classes: cardboard (393 images), glass (491), metal (400), paper (584), plastic (472), and trash (127). These categories cover a wide range of common recyclable and non-recyclable waste.

An exploratory analysis of the dataset reveals a moderate class imbalance, particularly with the "trash" class being underrepresented. This imbalance highlights the importance of applying data augmentation techniques to prevent biased learning.

In addition, there is significant intra-class variation—for example, plastic items differ in shape, color, and opacity—while inter-class similarities (*e.g.*, between cardboard and paper) make the classification task more challenging. The dataset also includes considerable background variation—some images have plain or uniform backgrounds, while others include complex or cluttered scenes. This diversity poses realistic challenges for the classification models and justifies the use of robust augmentation techniques and ensemble learning.

## GANS for data augmentation

GANs have the potential to understand the underlying patterns of waste object images and generate synthetic samples that closely resemble real images. This capability expands the dataset used in training and facilitates the training of more robust and diverse models for waste object detection and classification. Augmenting the dataset with synthetic images generated through GANs makes the models resilient to changes in the appearance of waste objects, background clutter, and lighting conditions (*Bowles et al., 2018*). It enables the model to better generalize to unseen data and provides a significant performance boost for waste management scenarios.

They are made up of two different models, a discriminator and a generator. In this case, the discriminator will attempt to determine if the image is real or a fake image that was

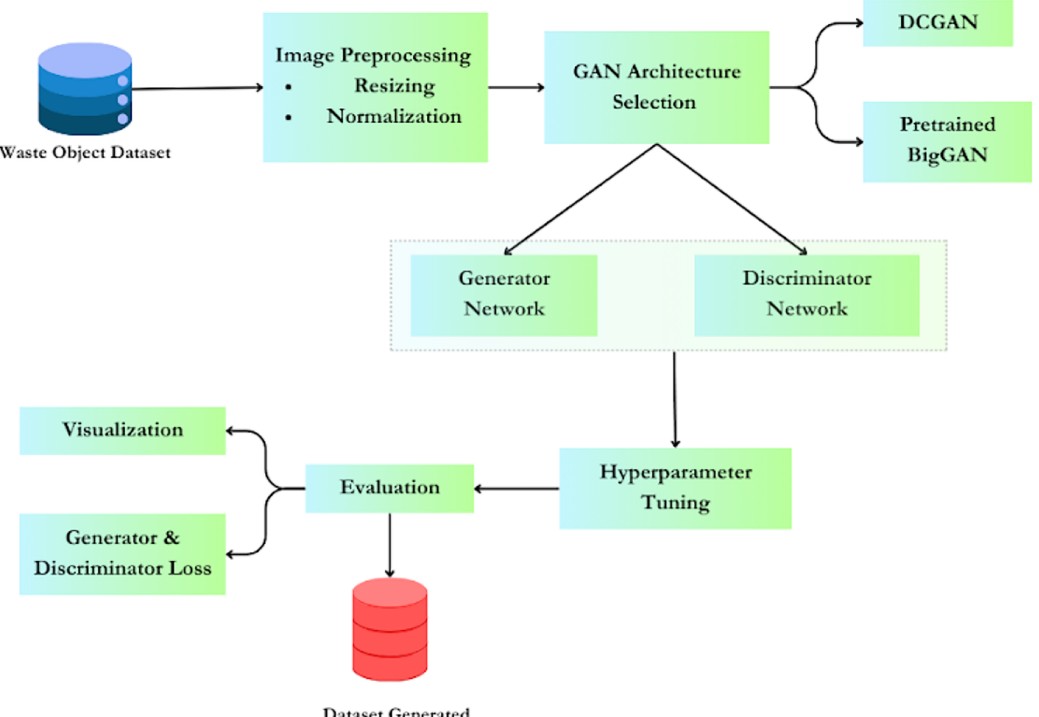

**Figure 1 Methodology for using GAN for data augmentation.**

provided by the generator, while the generator will attempt to produce synthetic or fake images that match the training images. Throughout training, the discriminator aims to improve at correctly identifying and classifying phony images, while the generator constantly strives to produce more convincing fake images than the discriminator. When the generator creates exact fakes that exactly resemble training data, an equilibrium is reached where the discriminator is compelled to detect, at 50% accuracy, whether the output generated is real or fake (*Lu et al., 2022*).

## GAN architecture design

Figure 1 explains the proposed methodology, outlining the process of working with generative adversarial networks (GANs) for data augmentation of waste object images, with a specific focus on deep convolutional generative adversarial networks (DCGAN) and pretrained BIGGAN models. A typical architecture of GAN consists of two models: a generator and a discriminator. A typical GAN architecture consists of two models: a generator and a discriminator. The generator is tasked with generating samples that closely mimic the training images (referred to as 'fake' images). Conversely, the discriminator ensures that the generated images closely resemble either the training image set or are altered versions of the original images. During the training process, the generator aims to deceive the discriminator classifier by producing fake images that appear increasingly realistic. Simultaneously, the discriminator endeavours to improve its detection rate to

discern fake images from real ones. Eventually, the generator becomes adept at generating high-quality fake images that are indistinguishable from real ones, even to the human eye. At this stage, the discriminator is challenged to accurately determine whether the images output by the generator are fake or real (*Erabati & Araujo, 2024*).

Consider *x* which denotes data representing an image. D(x), the discrimination network. There is no difference in the self-consistency loss, and it can be associated with the direct probability of x, that is, from the training set, and not from the generator. Because the model is working with images, x being D's input is an image of trained community workers size $64 \times 64 \times 3$ (*CCHANG, 2018*). Common sense dictates the D(x) to be almost always HIGH when x belongs to training data and almost always LOW when x is from the generator. By means of this function, the activation grid is being converted into a binary classifier, working on the principle of binary classifier.

Consider the generator's notation, denote *z* be a latent space vector representation that is sampled from a normal distribution. *G(z)* denotes the generator function which takes z as input. The job of G is to learn the distribution that the training data are distributed from ($p_{data}$), so it can produce new fake samples from that learned distribution of latent space representation. Equation (1) shows the main objective of GAN.

The loss function for GAN is stated below:

$$minmax\ V(D, G) = E_{x \sim pdata(x)}\left[\log D(x) + E_{z \sim pz(z)}[log(1 - D(G(z)))]\right]. \tag{1}$$

## Training procedure

Let 'x' represent an image's data. D(x) stands for the discriminator network, from which we get the returns in the form of scalar probability of the data label "x," *i.e.*, from data that is designated as training data rather than generating data. Since we are working with picture data, the image supplied as input to D(X) will have the CHW size $3 \times 64 \times 64$ (*Li et al., 2022*). D(x) must, in essence, be LOW when x is the generator's output and HIGH when x comes from the training set.

The latent space vector for generator, which was taken from a typical normal distribution, is represented by the symbol z. The generator function, represented by G(z), takes a latent vector z as input and outputs data space. Here, G creates fictitious samples from those learnt distributions while attempting to comprehend the distribution from which the training data (p data) are dispersed.

*Weight Initialization:* every model should have its weight initialized using a normal distribution with mean = 0 and stdev = 0.02, according to DCGAN guidelines. The function weights_init is tasked with taking an initialized model as input and re-initializing all batch normalization, convolutional, and transposition layers to satisfy this requirement.

*Generator:* the latent space vector serves as the generator's input, which it maps to data-space. Given that our data are photos, creating an RGB image of the same dimensions ($3 \times 64 \times 64$) as the training images—makes sense. Actually, four two-dimensional convolutional transpose layers are employed in the series succeeded by a relu activation, a

2d batch norm layer, and a 2d batch norm layer. The generator's output is passed *via* a tanh function in order to obtain the final output in the input data domain of [−1, 1] (*Poole et al., 2016*). Given that the batch norm functions in the DCGAN article come after the convtranspose layers, a few observations are necessary. These layers facilitate gradient flow during training.

Discriminator: an image as input, the model generates then a train of numbers referring to the probability that the input image is real as opposed to fake which is indicated by the scalar probability. This is called a binary classification network discriminator D. In this example, D is basically processing a $3 \times 64 \times 64$ input image by a sequence of layers which contain batchNorm2d, Conv2d, and LeakyReLU before senior probability is generated using Sigmoid activation function (*Yang et al., 2022*). Thus, constructing the architecture based on these parameters is beneficial to obtain the required solution as well as for complicated problems if expanding the layers are necessary, just BatchNorm, LeakyReLUs, the stridden convolution, and lots more. Unlike pooling, limited convolution is capable of building its own pooling function as contrasted to standard pooling that makes it a better instrumentality toward down sampling, according to the DCGAN research.

Loss functions and optimizers: after setting up the discriminator and generator, the learning can be monitored with loss functions and optimizers. Here, the loss function used is *BinaryCrossEntropy* defined in Eqs. (2) and (3) as:

$$\ell(x, y) = L = \{l_1, \ldots, l_N\}^\top \tag{2}$$
$$l_n = -[y_n \cdot \log x_n + (1 - y_n) \cdot \log(1 - x_n)]. \tag{3}$$

Part 1: Train the discriminator

Algorithm has to train the discriminator to its maximum classification accuracy of both real and fake data inputs. Here we try to maximize the *log(D(x))+log(1−D(G(z)))*.

A batch of real samples from the training set is generated and it is forwarded through D. After this, the loss *((x))* is calculated and then gradients with another backward pass are calculated. Secondly, we will develop samples with the current generator to form then a batch. This batch will then be forwarded through D, the loss will then be calculated (log(1 −D(G(z)))), and the accumulation of the gradients will be carried out with the backward pass thereafter. Each fake sample becomes a synthesis of the gradients from all the real and fake photos.

The GAN was trained on the training subset of the original dataset. After training stabilization, we generated 300 synthetic images for each underrepresented class (trash, cardboard, and metal) and 100–150 images for other classes to maintain proportional class distribution.

This approach brought the total number of images per class closer to balance, with the final training set containing both real and synthetic data. The generated images were visually inspected for quality and diversity, ensuring realistic samples that aligned with their class labels.

**Table 2 Input parameters for training statistics.**

| | | |
|---|---|---|
| Discriminator loss = loss of real + sum of fake | Loss_D | $log(D(x)+log(1-D(G(z))))$ |
| Generator loss | Loss_G | $log(D(G(z)))$ |
| Discriminator's average output for all real batch | D(x) | 1 and converges to 0.5 for better G |
| Discriminator's average output for all fake batch | D(G(z)) | 0 and converges to 0.5 for better G |

Part 2: Train the generator

The generator should be trained in such a way that the term $log(1-D(G(z)))$ is minimized. This helps to generate better and more realistic samples. This concludes that we maximize $log(D(G(z)))$. We do that by conditioning the generator output to be classified with the discriminator, which calculates G's loss as Ground Truth (GT) that uses real labels; then, computes G's gradients using a backward pass and finally update G's parameters with an optimizer step.

The training statistics are reported in Table 2. A fixed batch of noise is pushed after the conclusion of each epoch through the generator. With this, the progress of the generator's training can be visually tracked.

## Classification models used

### *Classification of waste objects*

In the study, for waste classification various pre-trained CNN models are used. Various steps like data preparation followed by data processing, model selection, and fine-tuning are done to evaluate the results. Efficient waste object classification enables efficient waste management and recycling processes. Proper waste material identification and sorting will help municipalities and facilities automate the sortation process, thus improving the effective recovery of resources with less landfill waste. The proposed methodology is depicted in Fig. 2. The further sections discuss the steps employed in the process of classifying the waste objects. Of more importance, it allows for specific recycling efforts and aids in proper waste disposal practices in terms of environmental sustenance.

## Data preparation and processing

Data preparation and preprocessing play a key role in ensuring effective and efficient waste classification models. First, all images are resized to standard resolution to ensure the dataset is compatible. This aids the computations in being more effective in the process for training the model, and secondly, this prevents the variations of image resolution that can actually hinder the performance of the model. This goes with the scaling and normalization of the pixel values of the images. This ensures that the pixel values will be within a given range, usually between 0 and 1. It aids in the stabilization of the training process and prevents any numerical instabilities. More importantly, normalization leads to more interpretable models and reduces the risk of overfitting (*Kulkarni et al., 2024*).

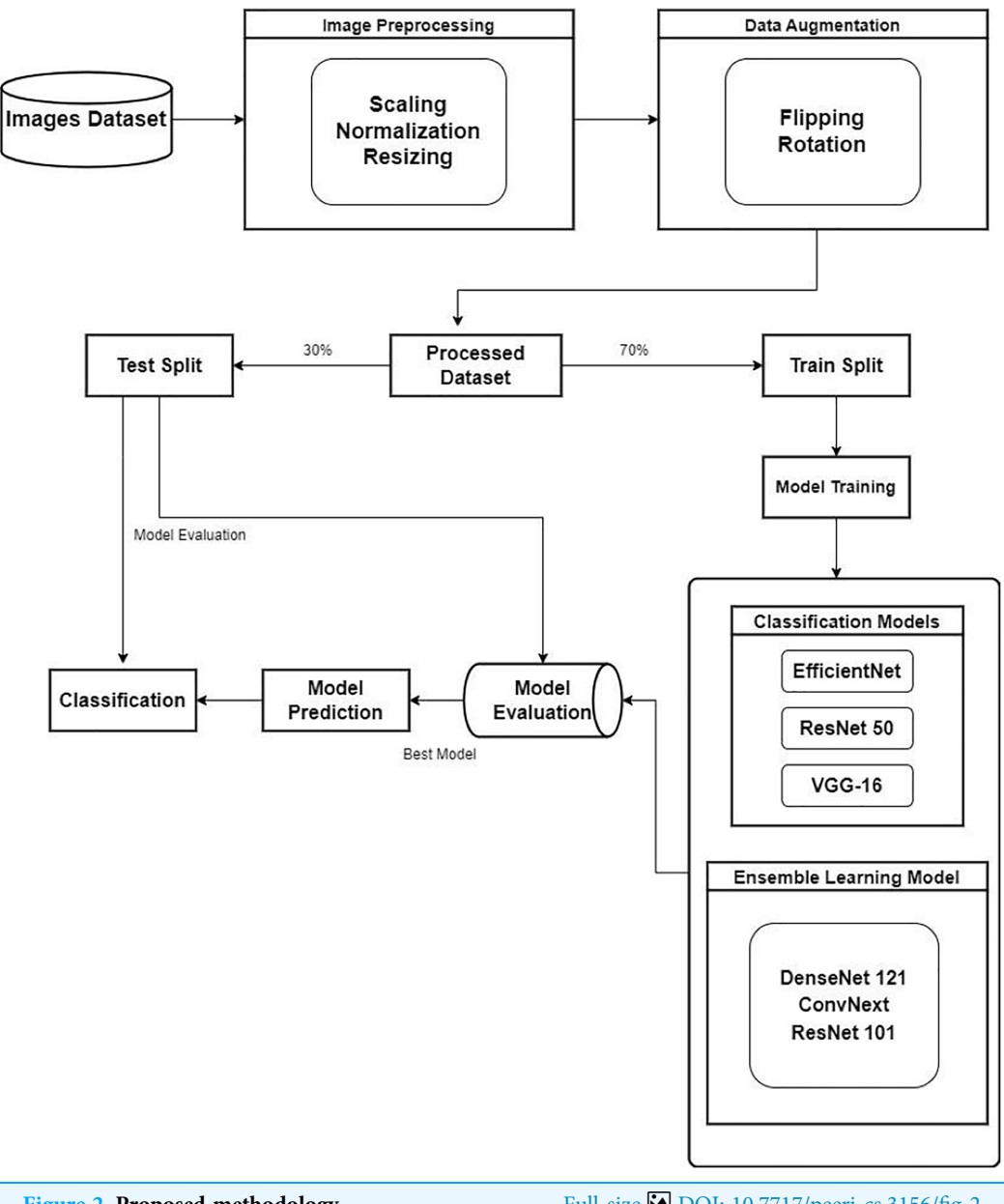

**Figure 2 Proposed methodology.**

These preprocessing steps constitute the backbone to good performance and generalization of the models. It standardizes the dimensions of the input and lets the models focus on learning meaningful features, being immune to variations in the size of images. The preprocessing of the data makes it possible for the model to be fed constantly scaled and centered inputs, which promotes stability and efficiency in training. This preprocessing approach facilitates the waste classification models' ability to identify various types of waste, thereby simplifying the adoption of effective waste management and recycling practices.

**Figure 3 Architecture of EfficientNet.**

## Classification models used

### EfficientNet

A group of convolutional neural network topologies called EfficientNet aims to accomplish great accuracy for image classification tasks and computational efficiency

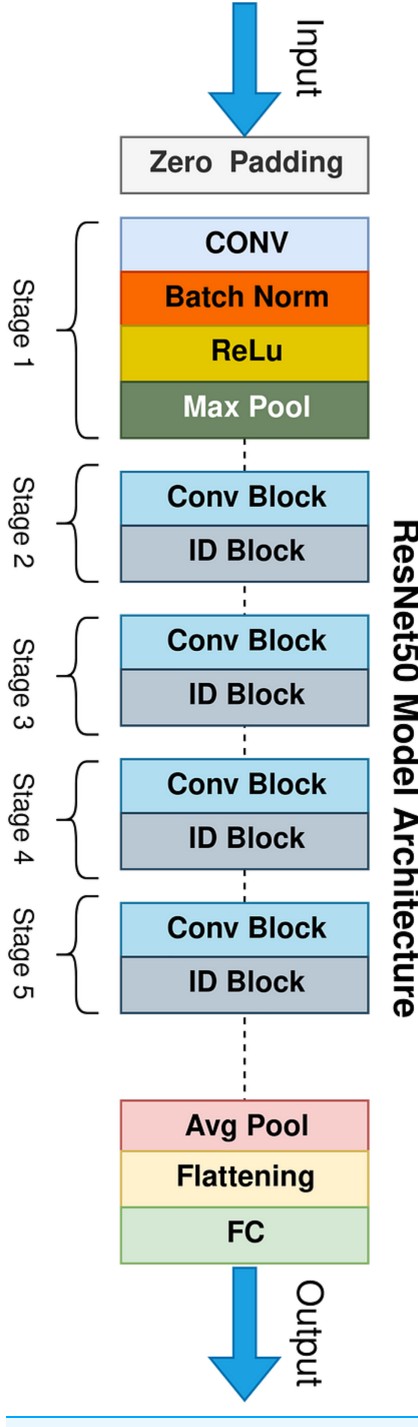

**Figure 4  Architecture of ResNet50.**     

(*Kulkarni et al., 2024*). State-of-the-art accuracy is attained on image classification tasks using substantially fewer parameters and computations than for other traditional CNN architectures such as VGG or ResNet. They are very convenient for deployment on resource constrained devices or in applications where faster inference time

is a must. The architecture of Efficient Net is described in Fig. 3 where the key components are as follows:

1. Base Network

a. Mobile Inverted Bottleneck Convolution (MBConv Layers): these are the basic blocks who balance the model complexity and performance. Two stages separate a standard convolution in depthwise separable convolutions: depthwise, where individual filters are used for applying them to each input channel, and pointwise, where all the outputs are combined using $1 \times 1$ convolutions.

b. Squeeze-and-Excitation Blocks: these are optimization modules that are based on dynamically recalibrating channel weights in the embeddings. It learns the weight of every feature channel. Therefore, it emphasizes the informative ones.

2. Compound Scaling

This is the first and main innovation of EfficientNet that solves the traditional tradeoff between model size, namely depth, width, and resolution, and performance by uniformly scaling all three dimensions, subject to fixed coefficients: α: layer number determining the count of layers in the network.

3. Channel Swish Activation

The activation function defined as *x * tanh(relu(x))*. This represents a smooth approximation of Rectified Linear Unit (ReLu), one of the power activation functions used in the neural network domain. It overcomes all of ReLU's computational drawbacks yet achieves almost the same performance.

### Resnet-50

ResNet-50 revolutionized deep learning by overcoming the vanishing gradient problem that haunted deeper neural networks (*Koonce & Koonce, 2021*). The residual blocks, as in Fig. 4 which aid in the network's learning from identity mappings, are the foundation of this—aside from the learned transformations. ResNet-50 is a kind of neural network that uses convolution where layers or residual blocks are stacked up. A common encoder-decoder structure is typical where the encoder comprises several convolutional layers, pooling layers (often max pooling), and residual blocks that progressively decrease the input image's spatial dimensions while extracting increasingly complex features. Decoder is typically not used in classification tasks like ResNet-50.

Residual block components typically consists of:

a. Convolutional layers: generally, $1 \times 1$ or $3 \times 3$ convolutional layers are used in ResNet-50 in any block.

b. Batch normalization: it stabilizes the training process, but normalizes the activations of each layer.

c. ReLU activation: this non-linear activation function adds a non-linearity in the network such that it can learn complex relationships between the inputs and output.

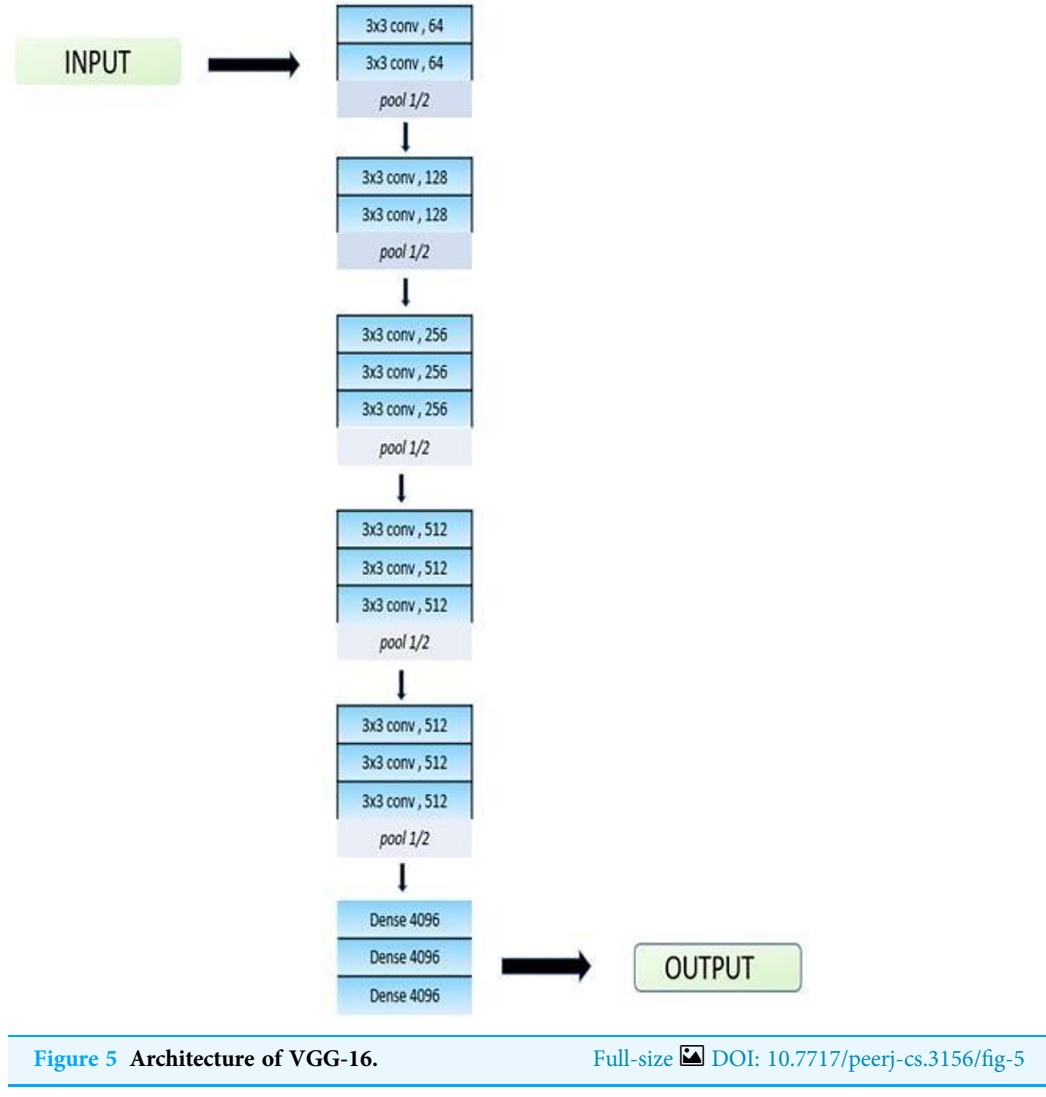

**Figure 5 Architecture of VGG-16.**

d. Shortcut connection: it feeds the input straight into the block's convolutional layers' output. This avoids the vanishing gradient issue and gives the network a shortcut to learn the identity mapping.

### VGG-16

As seen in Fig. 5, a convolutional neural network, in this case VGG-16, has a total of 16 layers. A pre-trained version of the network with over a million pictures under training is available through the ImageNet database. Its accuracy rate is 92.7%. VGG16 is unique as many hyper-parameters are replaced with max pool layer of 2 × 2 filters with the same padding and convolution layers of 3 × 3 filters with stride 1. The filters in VGG-16 model are used for learning different features and patterns in the input data to help the network make accurate predictions. In the neural network model, a stack of convolutional layers is

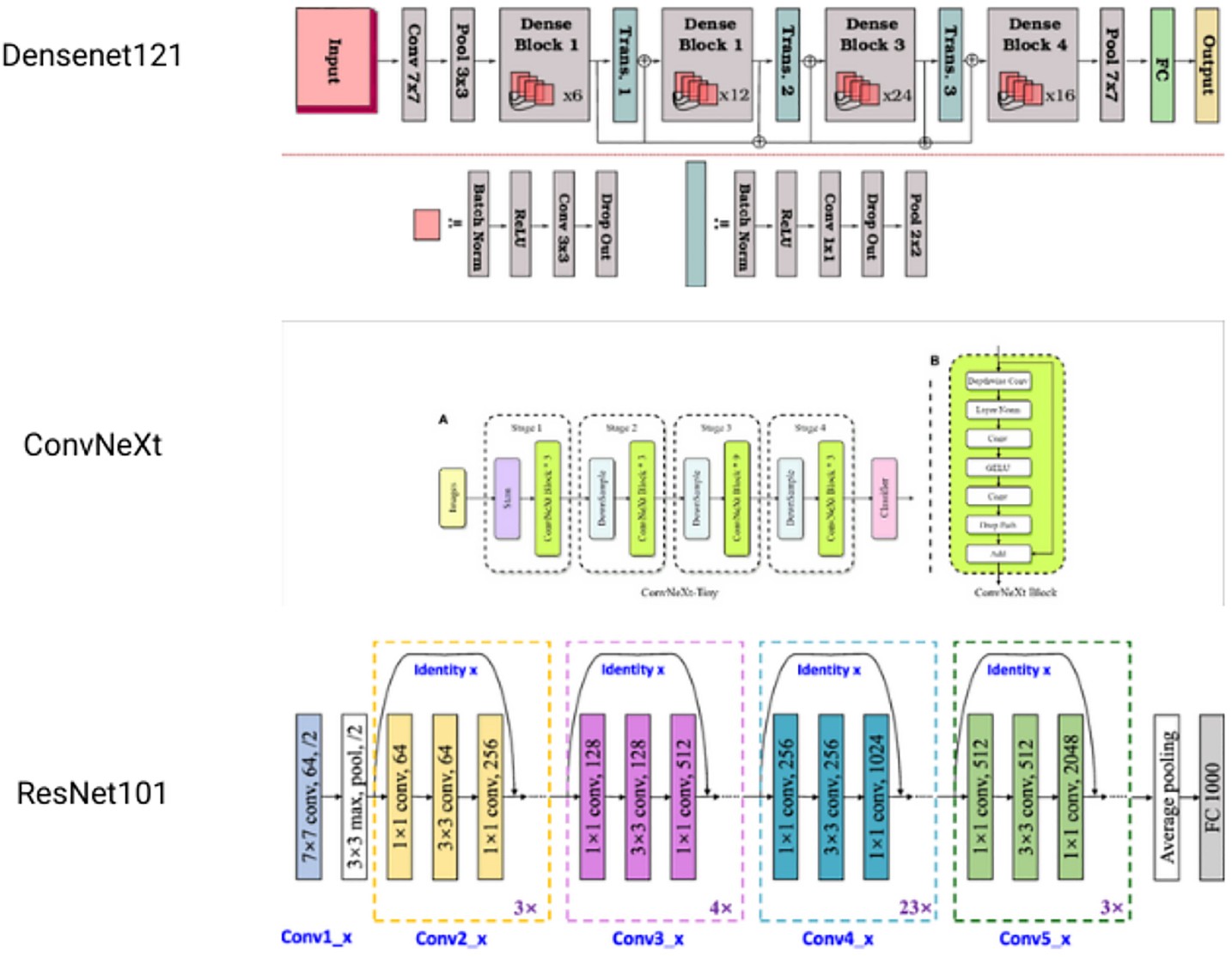

**Figure 6 Individual architectures of models used for ensemble.**

followed by three fully connected (FC) layers. The final categorization of the input data is carried out by fully-connected layers, which use the features that the convolutional layers have learned. The third requires 1,000 channels (one for each class) since it uses 1,000-way ILSVRC classification, while the previous two have 4,096 channels each. The final layer is the soft-max layer. One function used as an activation function is the Rectified Linear Unit (ReLU). ReLU decreases the chance of vanishing gradient and speeds up learning. Transfer learning is used to refine the model's upper layers.

The top layers of the model are fine-tuned using transfer learning so that the output can detect images from six classes. In the model, there is a dropout layer at 0.5. Dropout is used to avoid overfitting. The L2 regularization technique is also used in the model. The loss

function is enhanced by L2 regularization, which adds the sum of squares of the model parameters. This added term is proportional to the magnitude of the model parameters, so it encourages a model with very small weights. This results in a simpler and smoother model that is not overfit. The model is trained by using the ADAM optimizer. The optimizers are techniques or tools to minimize losses and adjust the parameters of your neural network, such as the learning rate and the weights. ADAM is the best optimizer as the training of the model consumes very little time and it is very efficient. A loss function for categorical cross entropy is used to train the model, and a SoftMax activation layer is used to reduce the bias.

## Ensemble learning model for waste object classification

Several basic models are integrated in the machine learning process, known as ensemble learning. These are combined to make a single best-predictive model. It is a powerful yet simple technique to improve final results. Here, a novel approach based on ensemble learning is suggested to strengthen the resilience and accuracy of our classification system. Figure 6 is the architecture of the models used. The methodology consists of the combination of three different CNN architectures: DenseNet121, ConvNext, and ResNet101. Each of them provides different features, allowing the usage of various feature representations and learning abilities (*Kanawade et al., 2023*).

*1. DenseNet121:* one of the dense connectivity features of Densely Connected Convolutional Network (DenseNet) is that is based on dense blocks having multiple convolutional layers. The key to feature propagation and information flow is effective feature propagation and information flow. The architecture of DenseNet121 comprises four dense blocks with a successive series of transition layers, which help to decrease the spatial dimensions and channel depth of feature maps. Following the last layers are a fully linked layer with a softmax activation for classification and a global average pooling.

*2. ConvNext:* ConvNext is a hybrid CNN architecture that incorporates both the convolutional and recurrent neural network units. Convolutional Long Short Term Memory (LSTM) units are introduced, which combine the convolutional operations with the memory capability of LSTM cells. The architecture of ConvNext consists of multiple convolutional layers in addition to convolutional LSTM units, allowing a model to capture both the spatial and temporal dependencies of the input data. The convolutional layers extract hierarchical features from the input images, and the convolutional LSTM units capture sequential patterns and long-range dependencies. The final layers usually consist of global pooling and fully connected layers for classification.

*3. Resnet101:* ResNet proposed the idea of residual connections that may help avoid the vanishing gradient problem in deep neural network architectures. ResNet101 is a type of ResNet, which has 101 layers. Residual blocks in ResNet 101 have both identity mapping and shortcut connections. The architecture of ResNet 101 consists of several groups of residual blocks, in which down sampling layers are inserted in each group. This allows for

the spatial reduction of feature maps. The last layers usually contain global average pooling and a classification layer by using fully connected layers.

At the inference stage, the three base models combined, and this was achieved by using a valid aggregation strategy, in which the strategies of averaging or weighted voting were used. This ensemble approach can mitigate the risk of overfitting and further improves the overall robustness of the classification system through the leveraging of the complementary strengths of each constituent model. Second, the ensemble model also showed higher robustness against variations in the input data and environmental conditions.

## System architecture

The computer infrastructure supporting this research is based on a robust hardware and software environment. The primary operating system utilized is Windows 11, providing a modern and efficient platform for development and execution. The hardware setup includes a system with 16 GB of RAM and a 512 GB SSD, ensuring ample memory and fast storage for handling large datasets of waste objects and running complex simulations. Additionally, the computational power is enhanced by a 4 GB Nvidia RTX 3050 GPU, which significantly accelerates tasks involving deep learning based GANs training and computer vision tasks like object detection. This combination of hardware and software components ensures a reliable and high-performance environment for conducting this research.

## RESULTS

This section presents the findings from experiments on data augmentation using generative adversarial networks (GANs) for waste object images. The goal of these experiments was to train a GAN model capable of generating realistic waste object images to supplement the limited training dataset, thereby enhancing the performance of waste object recognition systems. To evaluate the quality of the images generated by the DCGAN model, we relied on a combination of informal visual assessment and quantitative analysis. The visual inspection was conducted by the research team to subjectively assess the realism and coherence of the generated samples, though no formal human study was conducted. For quantitative evaluation, generator and discriminator loss trends over training epochs were analyzed. Binary cross-entropy was used as the loss function, offering insight into the model's learning behavior and stability. The training dynamics of the GAN model were assessed based on these loss curves to ensure convergence and consistency in generation quality. The generated images are visualized.

### Results of DCGAN

Figure 7A depicts the losses during the training of the generator and discriminator after 50 epochs. Following this, the epochs are increased to 100 where Fig. 7B plots the generated fake samples, and in Fig. 7B losses can be calculated with the graph. Furthermore, for a total of 500 epochs and then 1,000 epochs, the network becomes relatively better where the discriminator loss is seen to 0.0221 and that of the generator is 9.0897 which further decreases to 0.056 and 5.6738 for the discriminator and generator respectively. It can be

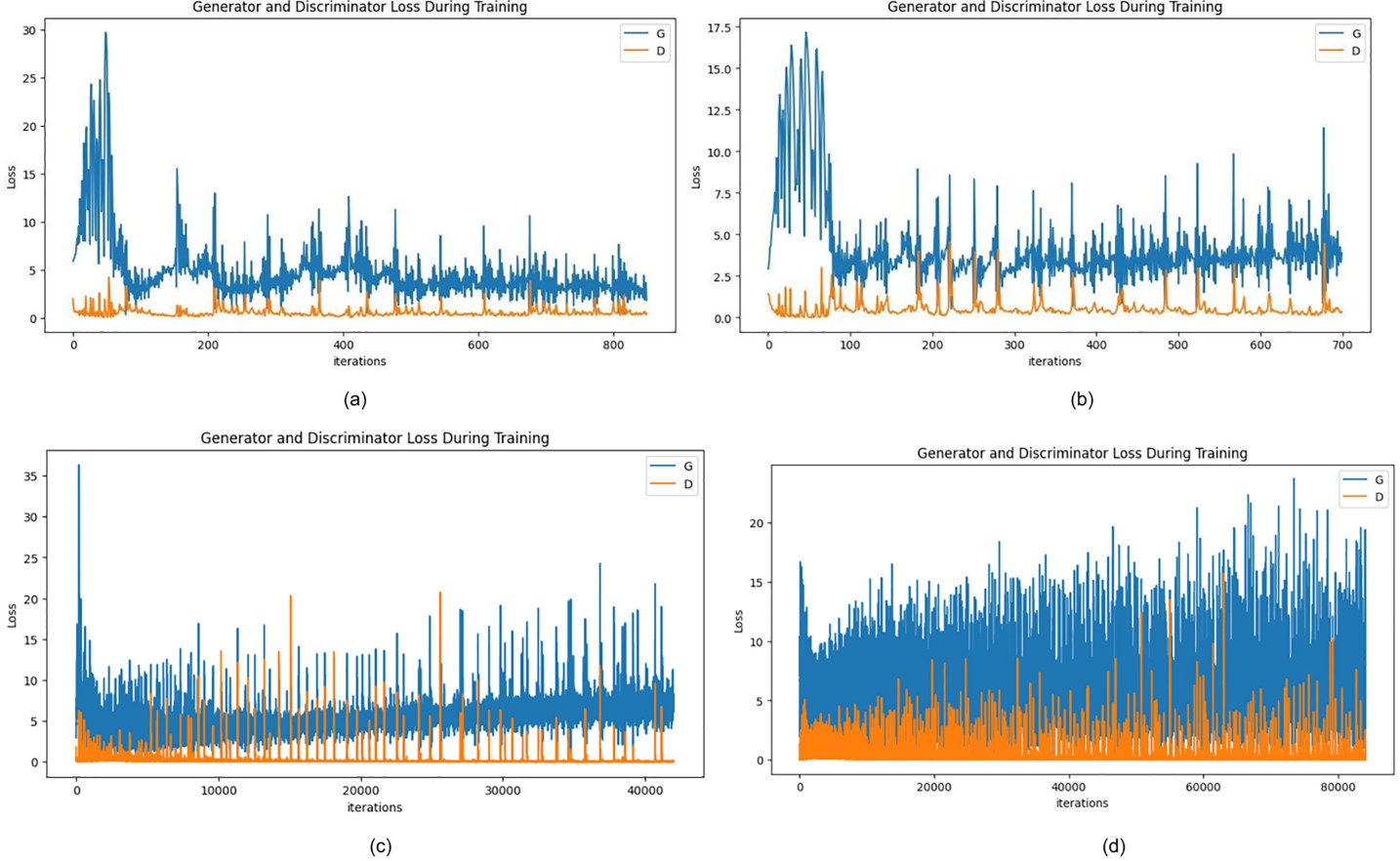

**Figure 7** (A) Generator and discriminator loss after 50 epochs, (B) Generator and discriminator loss after 100 epochs, (C) Generator and discriminator loss after 500 epochs, and (D) Generator and discriminator loss after 1,000 epochs.

seen, that both the generator and discriminator losses fluctuate with training, which means that the adversarial interactions between the generator and discriminator networks are in full action. The sample images generated are viewed in Figs. 8A–8D for more visualization.

Table 3 provides a detailed comparison of the convergence of the adversarial network where the individual network's losses are marked against the epochs. This is useful information about the dynamics of the adversarial training process and how the balance between the generator's performance for generating realistic images and the discriminator's performance for distinguishing between real and fake samples is gradually shifted from the changes in these losses over time. It provides much deeper insight into the learning dynamics inside the GAN framework and glimpses into the convergence of components of the network during training.

## Results of pretrained BIGGAN

Besides GAN-based augmentation, the study also utilizes the pre-trained BIGGAN model to synthesize high-fidelity waste object images. Figures 9, 10, 11 show a few synthesized images from the pre-trained BIGGAN model for three different waste object classes: plastic bottles, glass jars, and metal cans. These images are very realistic, with a lot of fine details

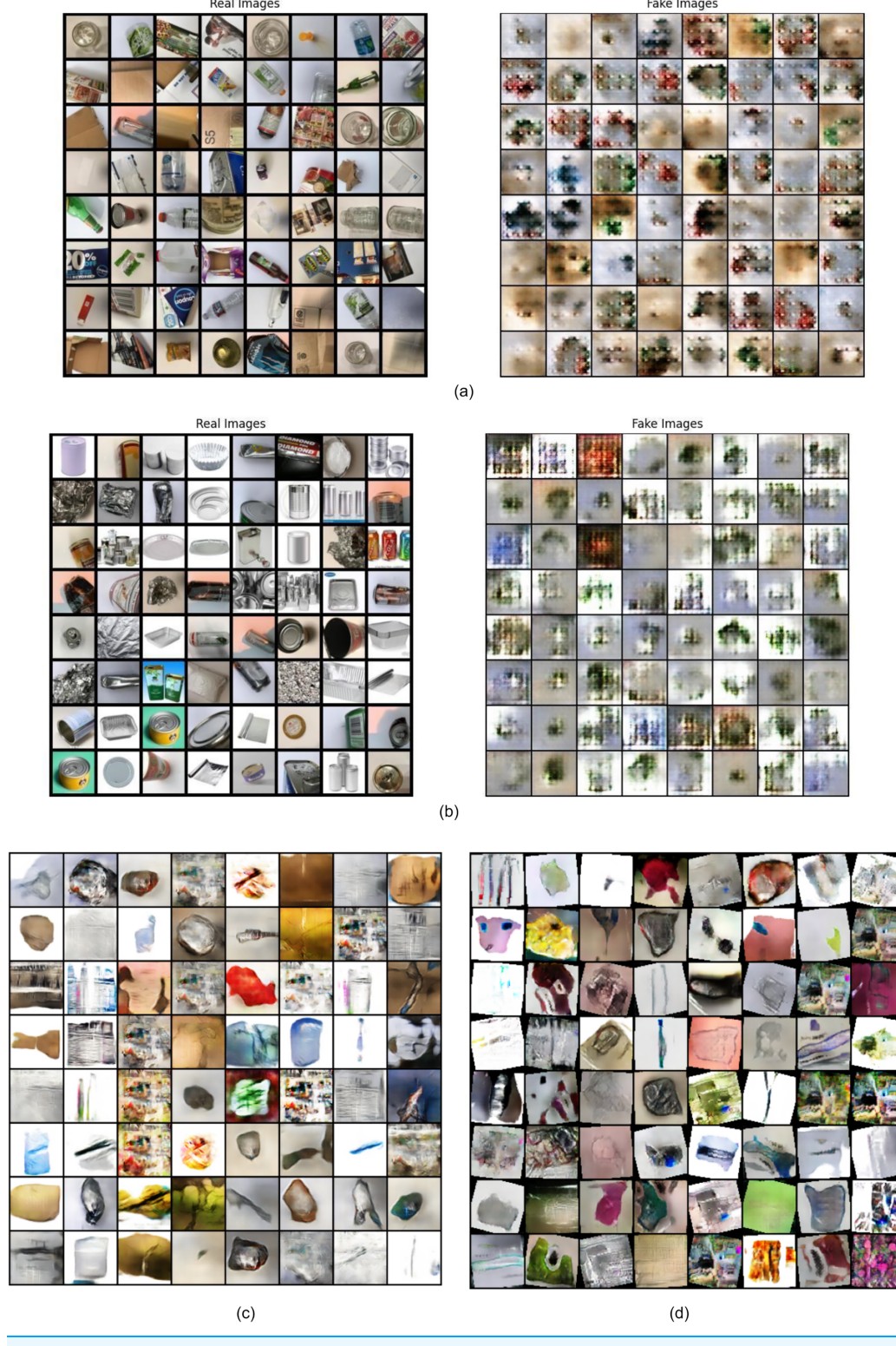

**Figure 8 (A)** Image generation after 50 epochs, **(B)** Image generation after 100 epochs, **(C)** Image generation after 500 epochs, and **(D)** Image generation after 1,000 epochs.

| Table 3 Report of training statistics. | | |
|---|---|---|
| **Epochs** | **Generator loss** | **Discriminator loss** |
| 5 | 14.6705 | 0.7637 |
| 50 | 4.2317 | 0.6637 |
| 100 | 2.5152 | 0.5028 |
| 500 | 9.0897 | 0.0221 |
| 1,000 | 5.6738 | 0.0506 |

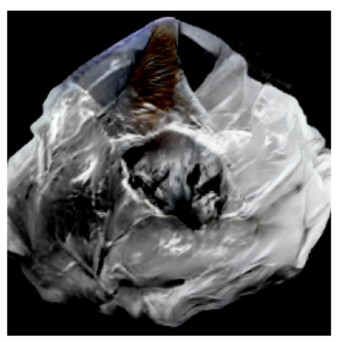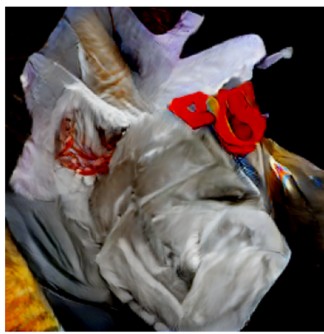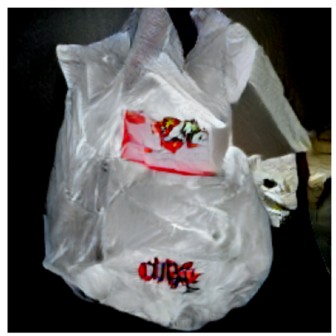

**Figure 9 Generated images for class plastic.**

visible in the intricate shapes and textures of each waste object class. The pre-trained BIGGAN model generates images with more fine details and more natural appearances because of its advanced architecture and large-scale training on many images. When we compare the images generated from DCGAN and the pretrained BIGGAN, the images generated by pretrained BIGGAN are of higher resolution as compared to DCGAN.

## Results for object detection

### Results for classification

Experiments have been carried out to determine the effectiveness of the classification models. The results of every classification model were inferred concerning the evaluation metric of "Accuracy".

Accuracy was used as the primary metric for evaluation that measures the proportion of correctly classified instances out of the total instances to provide a straightforward indication of model performance. It is calculated using Eq. (4).

$$Accuracy = \frac{Tp + Tn}{Tp + Tn + Fp + Fn}. \tag{4}$$

Here, Tp (True Positives) are the correctly classified positive instances and Tn (True Negatives) represents correctly classified negative instances. Fp (False Positives) are the negative instances incorrectly classified as positive and Fn (False Negatives) are the positive instances incorrectly classified as negative.

The training and validation accuracy of EfficientNet is displayed in Fig. 12A. The model shows impressive accuracies of 98.678% and 96.16in 4% training and testing respectively.

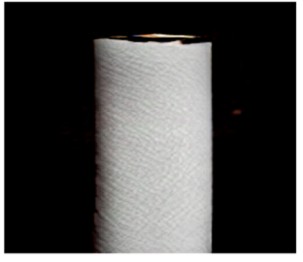 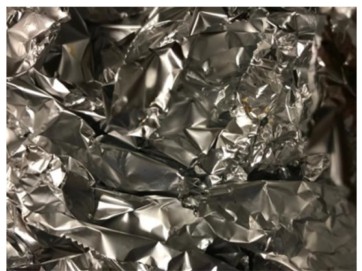 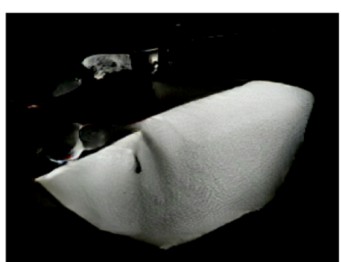

**Figure 10 Generated images for class paper.**

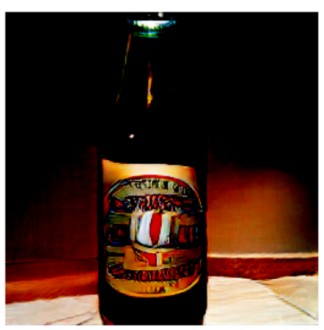 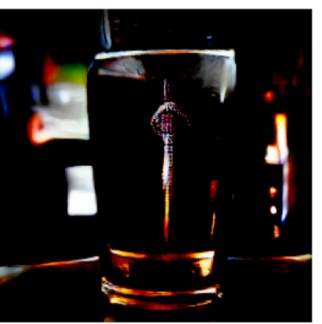 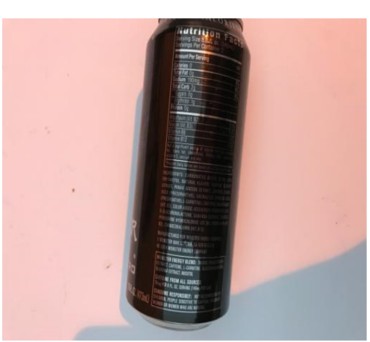

**Figure 11 Generated images for class glass.**

Figure 12B demonstrates the loss for the EfficientNet model. Similarly, in Fig. 13A, accuracies for the Resnet50 model are plotted against their epochs where the model has 77.981% on the train set and 70.344% on the test set. The graphs for loss during training and validation can be seen in Fig. 13B. The next model chosen for classification was VGG-16 which too performed relatively better than ResNet50. It exhibited an accuracy of 96.964% on training data and 90.458% on testing data. The accuracy for the graphs and loss are shown in Figs. 14A and 14B respectively.

### Ensemble learning model

With the proposed ensemble learning model, the accuracy was improved. It facilitated the model's transparency and interpretability by analyzing the performance of individual base models and their contribution to final predictions. By ensembling DenseNet121, ConvNext, and ResNet101, an accuracy of 99.99% was achieved while training. The test accuracy was found to be 99.80%. The graphs for the same are shown in Fig. 15. This implies that the fusion process proved to harness the collective intelligence of the base models and well adapt to the intricacies and nuances of the garbage dataset.

Table 4 presents the classification report highlighting the performance of the fusion process for every class along with the evaluation metrics like precision, recall, f1-score, and support. In Fig. 16, the confusion matrix is plotted to understand the predicted labels and true labels in the process of evaluating the model on the test dataset.

Table 5 is a comparison table for model variants used in the process of classification. The entire training of various models was done on 10 epochs with various

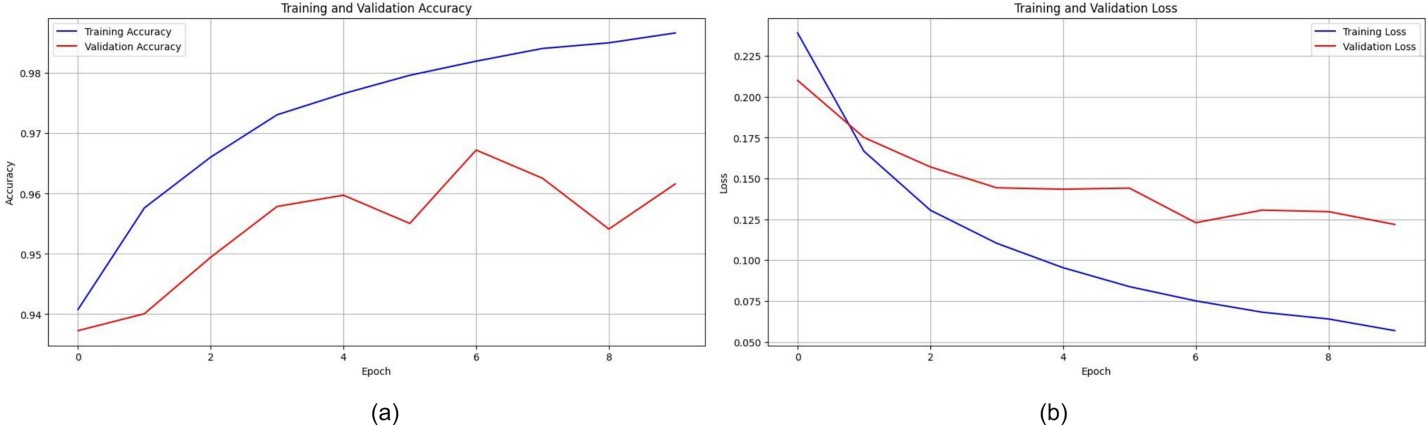

(a)                   (b)

**Figure 12** **(A)** Training and validation accuracy of efficient net, **(B)** training and validation loss of efficient net.

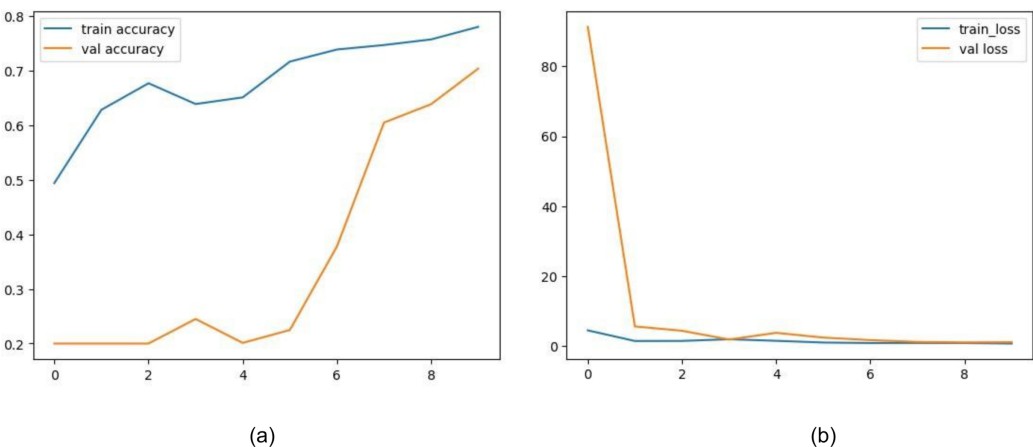

(a)                   (b)

**Figure 13** **(A)** Training and validation accuracy of ResNet50, **(B)** training and validation loss of ResNet50.

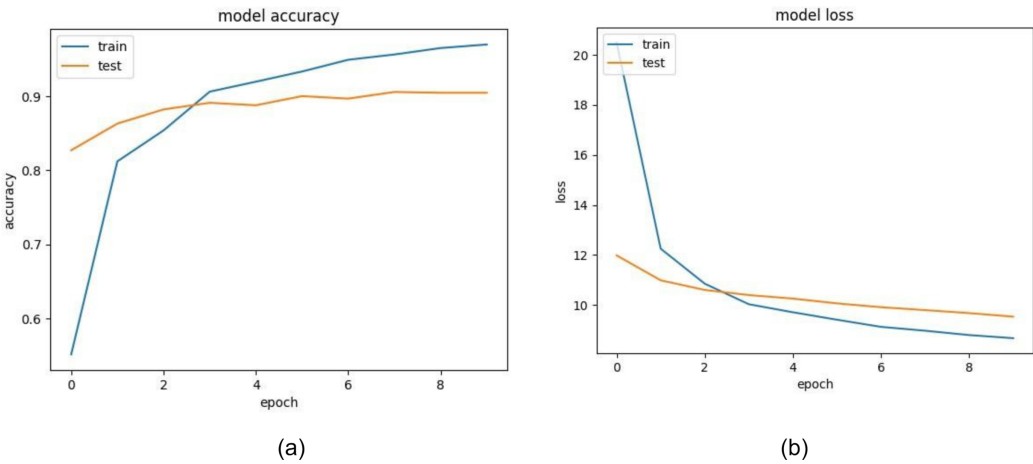

(a)                   (b)

**Figure 14** **(A)** Training and validation accuracy of VGG16, **(B)** training and validation loss of VGG16.

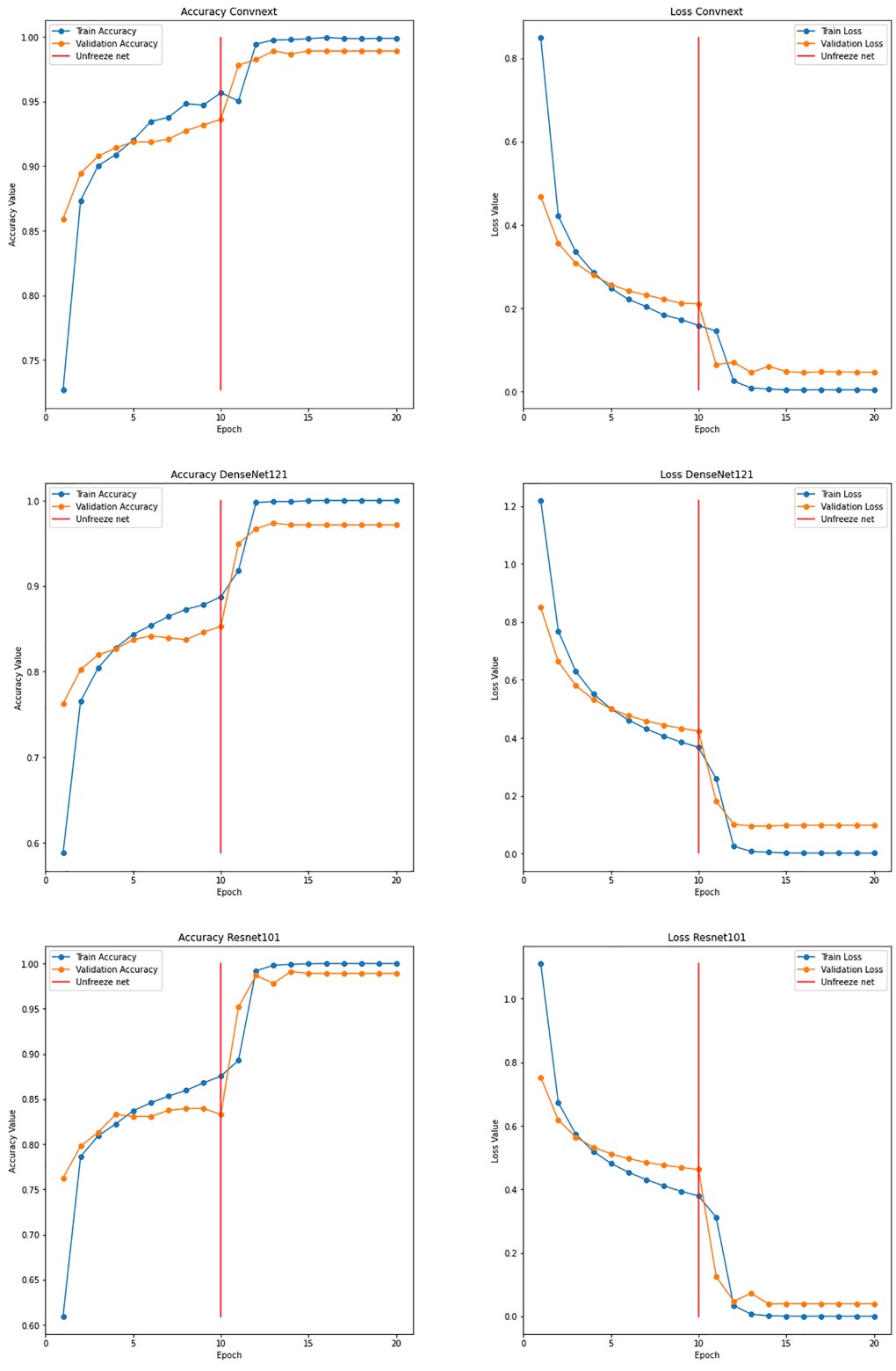

**Figure 15 Graphs depicting accuracy and loss for individual models used for ensemble learning.**

 

**Table 4 Classification report for ensemble mode.**

|  | Precision | Recall | F1-score | Support |
|---|---|---|---|---|
| Cardboard | 1.000000 | 1.000000 | 1.000000 | 64.000000 |
| Glass | 1.000000 | 1.000000 | 1.000000 | 105.000000 |
| Metal | 1.000000 | 1.000000 | 1.000000 | 86.000000 |
| Paper | 0.992126 | 1.000000 | 0.996047 | 126.000000 |
| Plastic | 1.000000 | 0.990741 | 0.995349 | 108.000000 |
| Trash | 1.000000 | 1.000000 | 1.000000 | 17.000000 |
| Accuracy | 0.998024 | 0.998024 | 0.998024 | 0.998024 |
| Macro avg | 0.998688 | 0.998457 | 0.998566 | 506.000000 |
| Weighted avg | 0.998039 | 0.998024 | 0.998023 | 506.000000 |

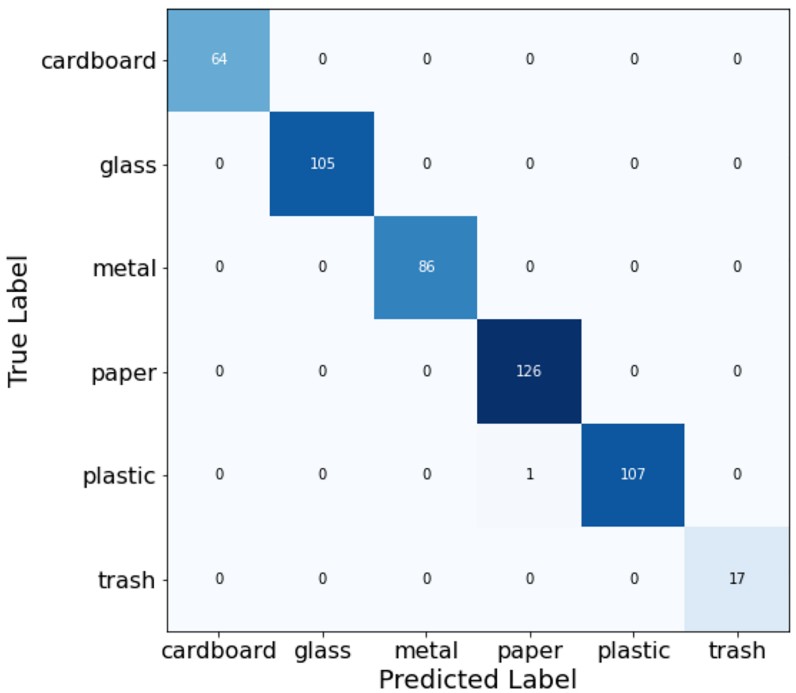

**Figure 16 Confusion matrix for ensemble classification model.**

**Table 5 Comparison of classification models used.**

| Models | Number of epochs | Training accuracies (%) | Testing accuracies (%) |
|---|---|---|---|
| Efficient Net | 10 | 98.678 | 96.164 |
| Resnet50 | 10 | 77.981 | 70.344 |
| VGG-16 | 10 | 96.964 | 90.458 |
| Ensemble model | 10 | 99.998 | 99.808 |

hyperparameters used in the process for better performance and generalization. Different hyperparameters were fine-tuned while the training process to find the right balance between the complexity of the model, the time needed for training, and the classification accuracy. The hyperparameters consists of the learning rate, the batch size, the choice of optimizer, weights, and dropout rate. In order to evaluate the model's capacity to generalize previously unknown data and make well-informed decisions about the selection of models and the makeup of the ensemble, standard evaluation measures such as accuracy, precision, recall, and F1-score were measured using a dataset for training and validation.

The ensemble model with DenseNet121, ConvNext, and ResNet101 achieved an accuracy of 99.99%, which is the best among the others. The resnet50 model gave less accuracy as the residual's calculation of the generated images and the images of datasets were a little different. Being an ensemble model, specifically the ResNet101 was selected, as the ResNet 50 model gave less accuracy, by adding this to the ensemble model enhanced the further accuracy.

## DISCUSSION

Environmental sustainability relies heavily on efficient waste management. Utilizing machine learning and AI-powered solutions shows great potential in improving the precision and speed of waste sorting and classification. Nevertheless, a major obstacle in this field is the absence of inclusive and well-rounded datasets necessary for developing strong classification models.

Most of the datasets on wastes are class-imbalanced, which bears low representations of certain wastes. This could result in biased models with poor performance on the less represented ones. GANs could alleviate this problem by generating synthetic images which will make the dataset balanced with underrepresented categories for a more balanced training set. This enhances model generalization across different kinds of wastes and, hence, better classification performance. In this regard, GANs can generate various realistic synthetic images of wastes, including the variations that did not exist in the original dataset. This diversity enables the model to recognize a wider range of waste objects under different conditions. Another challenge could be the high cost, both in money and time, associated with collecting and annotating large data sets of images of wastes. Synthetic data generation lowers the requirements of large real-world data collection. Thus, with GAN an effective waste classification model can be created. Most standard GAN architectures are ineffective as they do not capture the escalation in variations in more complex waste data. To cope with this, we made enhancements on the standard deep convolutional GAN (DCGAN) framework in a manner of learning how to generate fake samples that belong to underpopulated waste classes. These adaptations have included the modification of structures of the generator and discriminator models to enable it on capturing the details that relate to waste materials. Moreover, using a weighted loss function, we aimed at giving higher importance to less involved classes in order to have balanced data in the augmented set. To counteract the problems of traditional augmentation methods, the generated images were quantitatively

compared using perceptual quality metrics in order to confirm their similarity with real-world waste data.

This work is different from previous research where studies have used a single predictor model for classification and, as such, present an ensemble of DenseNet121, ConvNext, and ResNet101. Each model brings in different advantageous factors; DenseNet121 for feature reuse, ConvNext for computational advantages and ResNet101 for deep features. In inference a weighted voting mechanism is used to arrive at the final classification, where the confidence scores of the different models are combined. It also results in much improved generalisation and accuracy of the waste object classification compared to the single models alone. Improved classification accuracy means automated systems can more reliably distinguish between different types of waste. This leads to better sorting, reducing contamination in recycling streams and improving the overall effectiveness of waste processing. The detection of objects in complex scenarios for example within waste containers and dumpsites is a complicated affair. To overcome this issue, FasterRCNN with InceptionNetV2 has been employed for better feature extraction attributes than the former but lacks appropriate region proposal ideas. The detection framework received further fine-tuning trough transfer learning, where InceptionNetV2 was trained on the ImageNet and then fine-tuned on the waste dataset. This more advanced design is adaptable to changes in lighting and different environmental conditions, making it easier to efficiently identify waste objects.

Automated systems powered by enhanced classification models require less manual intervention, reducing labor costs. Moreover, accurate sorting minimizes waste processing errors, leading to cost savings in downstream recycling processes. By incorporating synthetic data to improve model performance, these systems can handle larger volumes of waste more efficiently, increasing throughput and operational efficiency.

## Limitations

Although GANs are very competent in generating high-quality images, the synthetic data still needs to be validated to ensure that it makes up a representative dataset without any kind of biases introduced in the model. The dataset's regional origin may limit the model's generalizability to waste from other locations with different packaging. Also, relying solely on RGB images restricts material identification to visual cues. Artificial intelligence (AI)-driven systems can be quite tricky to integrate with the pre-existing waste management infrastructure. Active knowledge about both technology and the operational processes of waste management is needed. While it demonstrates the potential of GAN-based augmentation with ensemble learning, several real-world challenges remain unaddressed. These include the need for geographically diverse data, handling of visually similar or mixed-material waste, frequent retraining due to evolving packaging, and integration into practical waste management workflows. While this study demonstrates the effectiveness of DCGAN-based augmentation combined with an ensemble of classification models, it does not evaluate the impact of alternative generative architectures (*e.g.*, StyleGAN, CycleGAN) or classifier configurations. It should weigh that the energy expenditure for training

BIGGAN models should atone for the environmental improvement linked with enhanced waste management.

## CONCLUSIONS

This research employed GANs to synthesize images of waste objects, thereby enhancing the accuracy of automatic waste classification systems. By increasing the size and diversity of the training dataset with synthetic images, models become more robust to real-world variations. The accuracy of automated waste management systems is often constrained by the limited availability of high-quality training data. To address this challenge, we augment our dataset with synthetic waste object images, thereby improving the robustness and accuracy of our classification model. Traditional waste management systems, which rely on manual sorting and outdated methods, struggle to handle the increasing complexity and volume of waste. Automated waste management systems have the potential to be more efficient, but their effectiveness is dependent on the availability of extensive and diverse training data. This article proposes a synthetic data augmentation scheme based on GANs, aimed at enabling machine learning models to learn from a broader variety of examples and to accurately identify and classify a diverse array of waste material. While this experiment has successfully automated the waste object detection and classification process through the use of DCGANs and several pre-trained CNN models, future work will focus on optimizing and fine-tuning data augmentation using GANs. Future research will investigate different GAN architectures, such as conditional GANs or progressive GANs, to generate more diverse and realistic synthetic waste object images. Additionally, exploring better ways to incorporate semantic information or context during the training phase of GANs could lead to even more effective data augmentation strategies. Fine-tuning pre-trained CNN models on waste-specific datasets will also enhance the performance of our object detection and classification models. Furthermore, integrating multimodal data sources, including depth information from depth sensors and contextual information from environmental sensors, can further improve the capabilities of our waste management system.

### Funding

This research is funded by the Centre for Advanced Modelling and Geospatial Information Systems (CAMGIS), Faculty of Engineering and Information Technology, the University of Technology Sydney, Australia. Moreover, supported by the National Research Foundation of Korea (NRF) grant funded by the Korea Government (MSIT) (No. NRF–2023R1A2C1007742), and Ongoing Research Funding Program (ORF-2025-14), King Saud University, Riyadh, Saudi Arabia. The funders had no role in study design, data collection and analysis, decision to publish, or preparation of the manuscript.

## Grant Disclosures

The following grant information was disclosed by the authors:

Centre for Advanced Modelling and Geospatial Information Systems (CAMGIS).

Faculty of Engineering and Information Technology, the University of Technology Sydney, Australia.

National Research Foundation of Korea (NRF).

Korea Government (MSIT): NRF–2023R1A2C1007742.

Ongoing Research Funding Program: ORF-2025-14.

King Saud University, Riyadh, Saudi Arabia.

## Competing Interests

The authors declare that they have no competing interests.

## Author Contributions

- Yashashree Mahale conceived and designed the experiments, performed the experiments, analyzed the data, performed the computation work, prepared figures and/or tables, and approved the final draft.
- Nida Khan conceived and designed the experiments, performed the experiments, analyzed the data, performed the computation work, prepared figures and/or tables, and approved the final draft.
- Kunal Kulkarni conceived and designed the experiments, performed the experiments, analyzed the data, performed the computation work, prepared figures and/or tables, and approved the final draft.
- Shilpa Gite conceived and designed the experiments, prepared figures and/or tables, authored or reviewed drafts of the article, and approved the final draft.
- Biswajeet Pradhan conceived and designed the experiments, prepared figures and/or tables, authored or reviewed drafts of the article, and approved the final draft.
- Abdullah Alamri conceived and designed the experiments, prepared figures and/or tables, authored or reviewed drafts of the article, and approved the final draft.
- Chang-Wook Lee conceived and designed the experiments, prepared figures and/or tables, authored or reviewed drafts of the article, and approved the final draft.
- Nandhini K. conceived and designed the experiments, prepared figures and/or tables, and approved the final draft.
- Mrinal Bachute conceived and designed the experiments, prepared figures and/or tables, and approved the final draft.

## Data Availability

The raw dataset and computer code are available at GitHub:

- https://github.com/Yashashree125/GANs-for-Data-Augmentation. (https://doi.org/10.5281/zenodo.16750079).

The dataset is available at Kaggle: CCHANG. (2018). Garbage Classification [Data set]. Kaggle. https://doi.org/10.34740/KAGGLE/DS/81794.

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
