# Peer review of "A comprehensive approach for waste management with GAN-augmented classification"

_PeerJ Computer Science, doi:10.7717/peerj-cs.3156_

## Round 0.1 · original submission · Major Revisions

Dear authors,

You are advised to critically respond to all comments point by point when preparing an updated version of the manuscript and while preparing for the rebuttal letter. Please address all comments/suggestions provided by reviewers, considering that these should be added to the new version of the manuscript.

Kind regards,
PCoelho

Reviewer 1 ·

Basic reporting

The abstract provides a good overview of the study, clearly stating the objectives, methods and results. However, it could be improved by briefly outlining the study’s limitations or challenges.

The manuscript is generally well-organized, with a logical progression from the introduction to methodology, results, and discussion. However, some sections, like the Related Work, seem quite lengthy and could benefit from more concise summaries.

Experimental design

The methodology lacks innovation, as it merely combines existing approaches. It then applies these methods to the problem.

The primary limitation of the experimental design lies in its focus on specific deep learning models (such as ResNet and DenseNet), which, although highly effective in many vision tasks, did not consider other potentially suitable models for waste classification, such as more lightweight architectures. Additionally, the paper does not provide a sufficient quantitative evaluation of the quality of the generated images, relying mostly on visual inspection, which may affect the objectivity of the experimental results to some extent.

Validity of the findings

The article dedicates a big portion to introducing existing models, which are already well-known to the readers. More focus could be placed on interpreting the results.

The article uses metrics such as accuracy and loss functions to evaluate model performance, which are common and reasonable in the field of deep learning. However, while the results show notable classification accuracy, other key evaluation metrics such as confusion matrix, precision, recall, and F1-score are missing, which are essential for providing a more comprehensive assessment of the model's performance, especially in cases of imbalanced classification data.

Additional comments

No Comments

Cite this review as

Reviewer 2 ·

Basic reporting

1. The article is devoted to the important topic of waste recognition in images using deep learning methods, as well as augmentation of existing datasets using generative adversarial neural networks.

2. The literature is not sufficiently referenced and relevant. The authors propose to augment datasets, but did not search for relevant datasets in sufficient detail, for example, a review of such datasets is available in the work:
- Yudin, D., et al. Hierarchical waste detection with weakly supervised segmentation in images from recycling plants. Engineering Applications of Artificial Intelligence, 128, p.107542. 2024

3. One of the key features of the paper is the augmentation of datasets using GAN models, while the same problem can be effectively solved using diffusion models, which is not mentioned in the paper, for example, a typical representative of this approach is the open source work PowerPaint:
- Zhuang, J., et al. A task is worth one word: Learning with task prompts for high-quality versatile image inpainting. arXiv preprint arXiv:2312.03594. ECCV 2024

4. In this regard, the Introduction is not adequately introduce the subject and make it clear what the motivation is. And the Related works section is also insufficient.

Experimental design

1. In the section of experiments on image classification in Table 6, only three well-known model architectures for image classification EfficientNet, ResNet, VGG-16 and ensembles based on them are considered. However, the authors ignore more modern classifier architectures based on backbones that have appeared in recent years: convolutional ResNeXt, ConvNext, transformer ViT, Swin, etc. In addition, for some reason, this Table 6 does not contain the results for the Densenet121, ResNet101 models shown in Fig 20.

2. The abstract of the article mentions "employing FasterRCNN + InceptionNet V2 for object detection in various environment scenes.", but the article does not contain quality metrics in the form of tables for various object detection methods. In addition, the successes of modern YOLO v8 - v11 models in this area are ignored. YOLO-based models achieve state-of-the-art indicators in this area.

Validity of the findings

1. One of the key shortcomings of the article is the lack of novelty of the proposed method. The work uses a combination of existing architectures and methods for training models. The authors did not clearly demonstrate their contribution and the originality of the methodology used.

2. The experiments and evaluations performed are not satisfactorily. There is a lack of a large number of experiments comparing the proposed approach with state-of-the-art approaches.

Additional comments

The work looks raw and requires significant revision.

Cite this review as

---

## Round 0.2 · Major Revisions

Dear authors,

After the previous revision round, some adjustments still need to be made. As a result, I once more suggest that you thoroughly follow the instructions provided by the reviewers to answer their inquiries clearly.

You are advised to critically respond to all comments point by point when preparing a new version of the manuscript and while preparing for the rebuttal letter. All the updates should be included in the new version of the manuscript.

Kind regards,
PCoelho

Reviewer 3 ·

Basic reporting

no comment

Experimental design

no comment

Validity of the findings

no comment

Additional comments

The article presents a proposition of adapting the DCGAN framework to focus on the synthesis of under-represented categories of waste, improving the number and quality of training data sets.
This work presents an effective GAN-based method that augments based on the data distribution and domain-specific characteristics, hence providing better generalisation. Additionally, the study also proposes an innovative method to form an ensemble of DenseNet121, ConvNext, and ResNet101 to achieve the optimised features for classification. In addition, the utilisation of Faster R-CNN for utilising InceptionNetV2 for object detections is made optimal for versatile environmental conditions to illustrate the flexibility of the proposed framework.
This research employed GANs to synthesise images of waste objects, thereby enhancing the accuracy of automatic waste classification systems. By increasing the size and diversity of the training dataset with synthetic images, models become more robust to real-world variations. The accuracy of automated waste management systems is often constrained by the limited availability of high-quality training data. To address this challenge, we augment our dataset with synthetic waste object images, thereby improving the robustness and accuracy of our classification model.

Cite this review as

Reviewer 4 ·

Basic reporting

* The state-of-the-art review is insufficient and should be extended. Specifically, in the field of waste classification using images under data scarcity, numerous relevant approaches exist. For instance, the following literature review provides a strong reference point to position the present work in the context of existing research: Arbeláez-Estrada, Juan Carlos, et al. "A systematic literature review of waste identification in automatic separation systems."
* The introduction contains multiple claims (lines 59–78) that are not adequately supported by references.
* If the study is exploratory (as mentioned in line 258), it should include an analysis of how variations in both the generative and classification models influence performance.
* The experimental section lacks an essential component in ML research: dataset exploration. Without data analysis (e.g., class distribution, brand variety, background complexity), it is difficult to assess the scope of the problem.
* The central contribution—employing GANs for data augmentation and assessing their influence on classifier performance—is insufficiently detailed. Key aspects such as the quantity of synthetic data used, which classes were augmented, whether the class imbalance was addressed, and how synthetic data affected performance are missing.

Experimental design

* Accuracy, the primary evaluation metric, is not suitable for imbalanced datasets.
* There is no comparison with alternative data augmentation techniques, which limits the evaluation of the proposed method’s effectiveness.
* In line 123, the authors state that the augmented data “closely mimics real-world variations, thereby expanding and diversifying the dataset.” However, there is no explanation or measurement of how this was verified.
* Numerous methods exist for synthetic data generation. The manuscript does not compare the proposed method against these alternatives.
* In the context of waste identification, the geographic location and source of the data are crucial, as packaging appearance varies regionally. The model relies solely on RGB images, making it dependent on visual cues, which introduces limitations in material identification.
* Known limitations of DCGANs—such as training instability, low image resolution, and the absence of conditional generation—are not addressed.
* In line 236, the authors critique existing datasets (TACO and TrashNet) for their lack of class diversity (e.g., medical or mixed materials). However, it is unclear how the proposed method can address this, especially considering the GAN is not conditionally trained to generate specific classes.
* The ensemble of three CNNs proposed is not novel and has been previously used in waste classification (e.g., Zheng, H.; Gu, Y. "Encnn-upmws: Waste classification by a CNN ensemble using the UPM weighting strategy.").
* In line 765, the authors claim the ensemble model improves transparency and interpretability. This claim is unsubstantiated, as ensembles typically reduce interpretability due to increased model complexity.

Validity of the findings

* The applicability of the method to real-world waste classification is questionable. Waste data requires frequent updates due to changing packaging, and GANs are known for being challenging to train and fine-tune.
* The study lacks novelty. No architectural changes, loss function improvements, or methodological innovations are presented. Similar approaches have already been reported in the literature, including:
* Fan, J.; Cui, L.; Fei, S. "Waste Detection System Based on Data Augmentation and YOLO_EC." Sensors, 2023.
* Kumsetty, N.V. et al. "An Approach for Waste Classification Using Data Augmentation and Transfer Learning Models."
* The integration of the method into an actual waste management workflow is not discussed. How can this be applied in practice? Would waste need to be manually separated beforehand to ensure one object per image?
* Key details are missing, such as:
* The volume of synthetic data used.
* The performance variation of the classifier with different proportions of synthetic data.
* Comparisons with other augmentation methods.
* Line 645 mentions human evaluation of GAN outputs, but the scale, criteria, and methodology for this evaluation are not described.

Additional comments

* The proposed data augmentation method using GANs has already been applied to various domains, including waste classification. The contribution does not offer new technical insights or methodological advances.
* Current challenges in waste classification via image analysis are not addressed, such as:
* The need for geographically localized datasets, as packaging differs by region.
* Packaging made from materials that visually resemble others.
* The frequent need to retrain models due to constantly evolving packaging designs.
* Integration into existing waste management workflows.
* Handling of complex waste types, such as mixed materials.
* Managing the broad diversity of waste beyond the main categories.
* If local data is required to capture packaging peculiarities, how does the proposed approach handle external datasets with different distributions? Can it generalize or adapt to such variations?
* A crucial missing component is the analysis of how the classifier's performance improves with increasing levels of synthetic data. At a minimum, this relationship should be reported, and ideally compared to other known augmentation techniques.
* It is unclear how the synthetic data was incorporated into training. Since the GAN is not class-conditional, were the generated images manually labeled before being used for training the classifier?
* Based on the results section, it appears the GAN and classifier were trained and evaluated independently. This undermines the purpose of using GAN-generated data to improve classification, as their integration is essential to validate the proposed approach.

Cite this review as

---

## Round 0.3 · accepted · Accept

Dear authors, we are pleased to verify that you meet the reviewer's valuable feedback to improve your research.

Thank you for considering PeerJ Computer Science and submitting your work.

Kind regards
PCoelho